# Genetic surveillance in the Greater Mekong subregion and South Asia to support malaria control and elimination

Christopher G Jacob[1], Nguyen Thuy-Nhien[2], Mayfong Mayxay[3,4,5], Richard J Maude[5,6,7], Huynh Hong Quang[8], Bouasy Hongvanthong[9], Viengxay Vanisaveth[9], Thang Ngo Duc[10], Huy Rekol[11], Rob van der Pluijm[5,6], Lorenz von Seidlein[5,6], Rick Fairhurst[12†], François Nosten[5,13], Md Amir Hossain[14], Naomi Park[1], Scott Goodwin[1], Pascal Ringwald[15], Keobouphaphone Chindavongsa[9], Paul Newton[3,5,6], Elizabeth Ashley[3,5], Sonexay Phalivong[3], Rapeephan Maude[6,16], Rithea Leang[11], Cheah Huch[11], Le Thanh Dong[17], Kim-Tuyen Nguyen[2], Tran Minh Nhat[2], Tran Tinh Hien[2], Hoa Nguyen[18], Nicole Zdrojewski[18], Sara Canavati[18], Abdullah Abu Sayeed[14], Didar Uddin[6], Caroline Buckee[7], Caterina I Fanello[5,6], Marie Onyamboko[19], Thomas Peto[5,6], Rupam Tripura[5,6], Chanaki Amaratunga[12‡§], Aung Myint Thu[5,13], Gilles Delmas[5,13], Jordi Landier[13,20], Daniel M Parker[13,21], Nguyen Hoang Chau[2], Dysoley Lek[11], Seila Suon[11], James Callery[5,6], Podjanee Jittamala[22], Borimas Hanboonkunupakarn[22], Sasithon Pukrittayakamee[22,23], Aung Pyae Phyo[5,24], Frank Smithuis[5,24], Khin Lin[25], Myo Thant[26], Tin Maung Hlaing[26], Parthasarathi Satpathi[27], Sanghamitra Satpathi[28], Prativa K Behera[28], Amar Tripura[29], Subrata Baidya[29], Neena Valecha[30], Anupkumar R Anvikar[30], Akhter Ul Islam[31], Abul Faiz[32], Chanon Kunasol[6], Eleanor Drury[1], Mihir Kekre[1], Mozam Ali[1], Katie Love[1], Shavanthi Rajatileka[1], Anna E Jeffreys[33], Kate Rowlands[33], Christina S Hubbart[33], Mehul Dhorda[5,6,34], Ranitha Vongpromek[6,34], Namfon Kotanan[22], Phrutsamon Wongnak[6], Jacob Almagro Garcia[35], Richard D Pearson[1,35], Cristina V Ariani[1], Thanat Chookajorn[22], Cinzia Malangone[1], T Nguyen[1], Jim Stalker[1], Ben Jeffery[35], Jonathan Keatley[1], Kimberly J Johnson[1,35], Dawn Muddyman[1], Xin Hui S Chan[5,6], John Sillitoe[1], Roberto Amato[1], Victoria Simpson[1,35], Sonia Gonçalves[1], Kirk Rockett[1,33], Nicholas P Day[5,6], Arjen M Dondorp[5,6], Dominic P Kwiatkowski[1,35], Olivo Miotto[1,6,35]*

*For correspondence: olivo@tropmedres.ac

Present address: †AstraZeneca, Gaithersburg, United States; ‡Mahidol-Oxford Tropical Medicine Research Unit, Mahidol University, Bangkok, Thailand; §Centre for Tropical Medicine and Global Health, University of Oxford, Oxford, United Kingdom

[1]Wellcome Sanger Institute, Hinxton, United Kingdom; [2]Oxford University Clinical Research Unit, Ho Chi Minh City, Viet Nam; [3]Lao-Oxford-Mahosot Hospital-Wellcome Research Unit (LOMWRU), Microbiology Laboratory, Mahosot Hospital, Vientiane, Lao People's Democratic Republic; [4]Institute of Research and Education Development (IRED), University of Health Sciences, Ministry of Health, Vientiane, Lao People's Democratic Republic; [5]Centre for Tropical Medicine and Global Health, University of Oxford, Oxford, United Kingdom; [6]Mahidol-Oxford Tropical Medicine Research Unit, Mahidol University, Bangkok, Thailand; [7]Harvard TH Chan School of Public Health, Harvard University, Boston, United States; [8]Institute of Malariology, Parasitology and Entomology (IMPE-QN), Quy Nhon, Viet Nam; [9]Centre of Malariology, Parasitology, and Entomology, Vientiane, Lao People's Democratic Republic; [10]National Institute of Malariology, Parasitology and Entomology (NIMPE), Hanoi, Viet Nam; [11]National Center for Parasitology, Entomology, and Malaria Control, Phnom Penh, Cambodia; [12]National Institute of

Allergy and Infectious Diseases, National Institutes of Health, Rockville, United States; [13]Shoklo Malaria Research Unit, Mae Sot, Thailand; [14]Chittagong Medical College Hospital, Chittagong, Bangladesh; [15]World Health Organization, Geneva, Switzerland; [16]Faculty of Medicine, Ramathibodi Hospital, Mahidol University, Bangkok, Thailand; [17]Institute of Malariology, Parasitology and Entomology (IMPEHCM), Ho Chi Minh City, Viet Nam; [18]Vysnova Partners Inc, Hanoi, Viet Nam; [19]Kinshasa School of Public Health, University of Kinshasa, Kinshasa, Democratic Republic of the Congo; [20]Aix-Marseille Université, INSERM, IRD, SESSTIM, Aix Marseille Institute of Public Health, ISSPAM, Marseille, France; [21]Susan and Henry Samueli College of Health Sciences, University of California, Irvine, Irvine, United States; [22]Faculty of Tropical Medicine, Mahidol University, Bangkok, Thailand; [23]The Royal Society of Thailand, Bangkok, Thailand; [24]Myanmar-Oxford Clinical Research Unit, Yangon, Myanmar; [25]Department of Medical Research, Pyin Oo Lwin, Myanmar; [26]Defence Services Medical Research Centre, Yangon, Myanmar; [27]Midnapore Medical College, Midnapur, India; [28]Ispat General Hospital, Rourkela, India; [29]Agartala Medical College, Agartala, India; [30]National Institute of Malaria Research, Indian Council of Medical Research, New Delhi, India; [31]Ramu Upazila Health Complex, Cox's Bazar, Bangladesh; [32]Malaria Research Group and Dev Care Foundation, Dhaka, Bangladesh; [33]Wellcome Trust Centre for Human Genetics, University of Oxford, Oxford, United Kingdom; [34]Worldwide Antimalarial Resistance Network (WWARN), Asia Regional Centre, Bangkok, Thailand; [35]MRC Centre for Genomics and Global Health, Big Data Institute, Oxford University, Oxford, United Kingdom

## Abstract

**Background:** National Malaria Control Programmes (NMCPs) currently make limited use of parasite genetic data. We have developed GenRe-Mekong, a platform for genetic surveillance of malaria in the Greater Mekong Subregion (GMS) that enables NMCPs to implement large-scale surveillance projects by integrating simple sample collection procedures in routine public health procedures.

**Methods:** Samples from symptomatic patients are processed by SpotMalaria, a high-throughput system that produces a comprehensive set of genotypes comprising several drug resistance markers, species markers and a genomic barcode. GenRe-Mekong delivers Genetic Report Cards, a compendium of genotypes and phenotype predictions used to map prevalence of resistance to multiple drugs.

**Results:** GenRe-Mekong has worked with NMCPs and research projects in eight countries, processing 9623 samples from clinical cases. Monitoring resistance markers has been valuable for tracking the rapid spread of parasites resistant to the dihydroartemisinin-piperaquine combination therapy. In Vietnam and Laos, GenRe-Mekong data have provided novel knowledge about the spread of these resistant strains into previously unaffected provinces, informing decision-making by NMCPs.

**Conclusions:** GenRe-Mekong provides detailed knowledge about drug resistance at a local level, and facilitates data sharing at a regional level, enabling cross-border resistance monitoring and providing the public health community with valuable insights. The project provides a rich open data resource to benefit the entire malaria community.

**Funding:** The GenRe-Mekong project is funded by the Bill and Melinda Gates Foundation (OPP11188166, OPP1204268). Genotyping and sequencing were funded by the Wellcome Trust (098051, 206194, 203141, 090770, 204911, 106698/B/14/Z) and Medical Research Council (G0600718). A proportion of samples were collected with the support of the UK Department for International Development (201900, M006212), and Intramural Research Program of the National Institute of Allergy and Infectious Diseases.

## Introduction

In low-income countries, particularly in sub-Saharan Africa, malaria continues to be a major cause of mortality, and intense efforts are underway to eliminate *Plasmodium falciparum* parasites, which cause the most severe form of the disease. However, *P. falciparum* has shown a remarkable ability to develop resistance to antimalarials, rendering therapies ineffective and frustrating control and elimination efforts. This problem is most acutely felt in the Greater Mekong Subregion (GMS), a region that has repeatedly been the origin of drug-resistant strains (*Dondorp et al., 2009*; *Noedl et al., 2008*; *Plowe, 2009*; *Roper et al., 2004*; *Mita et al., 2011*) and in neighboring countries including Bangladesh and India, where resistance could be imported. The GMS is a region of relatively low endemicity, with entomological inoculation rates 2–3 orders of magnitude lower than in Africa, where the vast majority of cases occur (*Chaumeau et al., 2018*; *Hay et al., 2000*). Infections are most common amongst individuals who work in or live near forests in remote rural parts of the region (*Cui et al., 2012*). Since infections are infrequent, a high proportion of individuals in this region are immunologically naïve, and develop symptoms that require treatment when infected. This results in high parasite exposure to drugs, which may be a major evolutionary driving force for the emergence of genetic factors that confer resistance to frontline therapies (*Escalante et al., 2009*). In the past, drug resistance alleles emerged in the GMS and subsequently spread to Africa multiple times, rolling back progress against the disease at the cost of many lives (*Mita et al., 2009*; *Trape et al., 1998*). Currently, global malaria control and elimination strategies depend on the efficacy of artemisinin combination therapies (ACTs) which are the frontline therapy of choice worldwide. Hence, in view of the emergence in the GMS of parasite strains resistant to artemisinin (*Dondorp et al., 2009*; *Ashley et al., 2014*; *MalariaGEN Plasmodium falciparum Community Project, 2016*) and its ACT partner drug piperaquine, (*Amaratunga et al., 2016*; *van der Pluijm et al., 2019*; *Leang et al., 2015*; *Spring et al., 2015*) the elimination of *P. falciparum* from this region has become a global health priority.

Elimination from the GMS presents significant challenges and, to ensure the most effective outcomes, NCMPs have to evaluate multiple changing factors: efficacy of frontline treatments, available alternatives, routes of spread, location of transmission hubs, importation of cases, and so on. In these assessments, NMCPs make extensive use of clinical and epidemiological data, such as those from routine clinical reporting and therapy efficacy studies. Parasite genetic data is less frequently available, and typically restricted to single genetic variants (*Ménard et al., 2016*), or small numbers of sites where quality sample collection protocols could be executed (*Lim et al., 2013*). However, routine mapping of a broad set resistance markers can keep NMCPs abreast of the spread of resistance strains, and help them predict changes in drug efficacy and assess alternative therapies, especially if dense geographical coverage allows mapping of resistance at province or district level. The increased affordability of high-throughput sequencing technologies now offers new opportunities for delivering such knowledge to public health, supporting the optimization of interventions where resources are limited (*Nagar et al., 2019*). Cost-effective implementation of genomic technologies, aimed at supporting public health decision-making, can make important contributions to malaria elimination (*Desmond-Hellmann, 2016*).

Here, we describe GenRe-Mekong, a genetic surveillance project conceived to provide public health experts in the GMS with timely and actionable knowledge, to support their decision-making in malaria elimination efforts. GenRe-Mekong analyzes small dried blood spots samples, which are easy to collect at public health facilities from patients with symptomatic malaria, and uses high-throughput technologies to extract large amounts of parasite genetic information from each sample. The results are captured in Genetic Report Cards (GRCs), datasets regularly delivered to NMCPs to keep them abreast of rapid epidemiological changes in the parasite population. The underlying technological platform is designed for low sample processing costs, promoting large-scale genetic epidemiology surveys with dense geographical coverage and large sample sizes.

To date, GenRe-Mekong has worked with NMCPs in Cambodia, Vietnam, Lao PDR (Laos), Thailand, and Bangladesh and has supported large-scale multisite research and elimination projects across the region (*van der Pluijm et al., 2019*; *von Seidlein et al., 2019*; *Chang et al., 2019*; *Landier et al., 2018*). The project has processed 9623 samples from eight countries, delivering data

to the 12 studies that submitted samples. In its initial phase, GenRe-Mekong has focused on applications relevant to the urgent problem of drug resistance. To facilitate integration into NMCP decision-making workflows, our analysis pipelines translate genotypes into predictions of drug resistance phenotypes, and present these as maps which are easily interpreted by public health officials with no prior training in genetics. In Laos and Vietnam, where GenRe-Mekong is implemented in dozens of public health facilities in endemic provinces, results from GenRe-Mekong have been used by NMCPs in assessments of frontline therapy options and resource allocation to combat drug resistance.

GenRe-Mekong protects individual patient privacy, while encouraging aggregation and sharing of standardized data across national borders to answer regional questions about epidemiology, gene flow, and parasite evolution (*Hamilton et al., 2019*). Aggregated data from multiple studies within GenRe-Mekong have powered large-scale genetic and clinical studies of resistance to dihydroartemisinin-piperaquine (DHA-PPQ), revealing a regional cross-border spread of specific strains (*van der Pluijm et al., 2019*; *Hamilton et al., 2019*). To power such high-resolution genetic epidemiology analyses of population structure and gene flow, GenRe-Mekong conducts whole-genome sequencing of selected high-quality samples, contributing to the open-access MalariaGEN Parasite Observatory (http://www.malariagen.net/resource/26) (*Pearson et al., 2019*). In this article, we summarize some key results from GenRe-Mekong, highlighting how they are used by public health officers to improve interventions. The data used in this paper are openly available, together with detailed methods documentation and details of partner studies, at http://www.malariagen.net/resource/29.

## Materials and methods

Additional detailed documentation on the methods used in this study is available from the article's Resource Page, at https://www.malariagen.net/resource/29.

### Sample collection

GenRe-Mekong samples were collected and contributed by independent studies with different goals, geographical coverage, and sampling strategies. Studies were managed by a local partner, such as a NMCP or a research organization, and often supported by a local technical partner. Most sampling sites were district or subdistrict health centres or provincial hospitals, selected by the local partner according to their public health or research needs. Each site was assigned a code, and its geographical coordinates recorded to support result mapping. GenRe-Mekong uses a common genetic surveillance study protocol covering the entire GMS, which can be locally adapted; this protocol was used for NMCP surveillance projects, after obtaining approval by a relevant local ethics review board and by the Oxford University Tropical Research Ethics Committee (OxTREC). Research studies included in their own protocol provisions for sample collection procedures, informed consent, patient privacy protection, and data sharing compatible with those in the GenRe-Mekong protocol, and obtained ethical approval from both a relevant local ethics review board, and their relevant institutional research ethics committee.

Samples were collected from patients of all ages diagnosed with *P. falciparum* malaria (including patients co-infected by other *Plasmodium* species) confirmed by positive rapid diagnostic test or blood smear microscopy. Participation in the study required written informed consent by patient, parent/guardian, or legally authorised representative (plus patient assent wherever required by national regulations), with the exception of Laos, where the Ministry of Health classified GenRe-Mekong as a surveillance activity for national benefit, requiring no additional informed consent. After obtaining consent, and before administering treatment, three 20 µL dried blood spots (DBS) on filter paper were obtained from each patient through finger-prick. GenRe-Mekong supplied study sites with kits containing all necessary materials, including strips of Whatman 31ET CHR filter paper, disposable lancet, 20 µl micropipette, cotton swab, alcohol pad, and plastic bag with silica gel for DBS storage. Scannable barcode stickers with unique identifiers were applied on the filter paper, the sample manifest where the collection date was recorded, and the site records. Samples were identified by means of these anonymous barcodes, and no patient-identifying information or clinical data were collected by GenRe-Mekong.

A number of participating studies also collected an optional anonymous questionnaire, to capture location of abode and work, occupation and travel history of the previous 2 months. These data are

intended for in-depth epidemiological studies, such as analyses of the contribution of travel to gene flow (*Chang et al., 2019*). Data from these questionnaires were stored in a separate system, and linked to genetic data by means of the tracking barcodes. They were not used in the present work.

## Sample preparation and genotyping

DBS samples were received and stored either at the Oxford University Clinical Research Unit, Ho Chi Minh City, Vietnam, or at the MORU/WWARN molecular laboratory, Bangkok, Thailand. Samples were registered and tracked in a secure bespoke online database, where location and date of collection were recorded. DNA was extracted from samples using high-throughput robotic equipment (Qiagen QIAsymphony) according to manufacturer's instructions. Extracted DNA was plated and shipped to the MalariaGEN Laboratory at the Wellcome Sanger Institute (WSI), Hinxton, UK, for genotyping and whole genome sequencing. Parasite DNA was amplified by applying selective whole genome amplification (sWGA) as previously described (*Oyola et al., 2016*).

Genotyping was performed by the SpotMalaria platform, described in the separate document 'SpotMalaria platform - Technical Notes and Methods' available from the Resource Page, which includes the complete list of genotyped variants and the details of the genotyping procedures for these variants. Briefly, the first version of SpotMalaria used multiplexed mass spectrometry arrays on the Agena MassArray system for typing most SNPs, and capillary sequencing for the artemisinin resistance domains of the *kelch13* gene. This was eventually replaced by an amplicon sequencing method, using Illumina sequencing of specific genome segments amplified by PCR reaction. The two implementations genotype a common set of variants, each iteration extending or improving on previous versions. Amplicon sequencing also offers greater portability, since it can be deployed on smaller sequencers in country-based laboratories.

## Genetic Report Cards generation

For each sample, genotypes were called for each variant analysed by SpotMalaria, and further processed to determine commonly recognized haplotypes associated with drug resistance (e.g. in genes *crt*, *dhfr*, *dhps*). Genetic barcodes were constructed by concatenating 101 SNP alleles. The generated genotypes, combined with sample metadata, were returned in tabular form to those partners who had submitted the samples along with explanatory documentation for the interpretation of the reports.

The genotypes generated were used to classify samples by their predicted resistance to different drugs. The prediction rules were based on the available data and current knowledge of resistance markers and are detailed in the separate document 'Mapping genetic markers to resistance status classification' available from the Resource Page. For each drug, samples were classified as 'sensitive', 'resistant', 'undetermined', or 'missing'– the latter identifying samples that failed to produce a valid genotype for the classification. Heterozygous samples, that is those containing genomes carrying both sensitive and resistant alleles, were classified as undetermined, due to lack of evidence for the drug resistance phenotype of such mixed infections.

In order to minimize the impact of call missingness, we also applied a set of *imputation rules* that predict missing alleles in the *crt*, *dhfr*, and *dhps* genes, based on statistically significant association with alleles at other positions. Associations were tested (using the threshold $p < 0.05$ by Fisher's exact test) using over 7000 samples in the MalariaGEN *Pf* Community Project Version 6 (*Pearson et al., 2019*). The rules for imputations were applied before phenotype prediction rules. They are detailed in the separate document 'Imputation of genotypes for markers of drug resistance' available from the Resource Page.

## Data aggregation and mapping of drug resistance

To estimate the frequency of resistant parasites for a given drug, we selected samples at the desired level of geographical aggregation (e.g. province/state or district), based on sampling location. After removing samples with missing and undetermined phenotype predictions for the desired drug, we counted the individuals predicted to be resistant ($n_r$) and sensitive ($n_s$), giving a total aggregation sample size $N=n_r+n_s$. Resistant parasite frequency was then computed as $f_r=n_r/N$. Maps of resistance frequency were produced using Tableau Desktop 2020.1.8 (RRID:SCR_013994, http://www.tableau.com/). To indicate levels of resistance, markers were colored with a custom green-orange-red

palette. Pie chart markers, used to represent allele proportions, were also derived from the same set of $N$ aggregated samples.

### Population structure analysis

Pairwise genetic distances between parasites were estimated by comparing genetic barcodes. To reduce error due to missingness, we first eliminated samples with more than 50% missing barcode genotypes; then we removed SNPs with missing calls in >20% of the remaining samples; and finally discarded samples with >25% missingness in the remaining SNPs. This produced a dataset of 87-SNP barcodes for 7490 samples from which genetic distances were estimated. For each sample $s$, we assigned a within-sample non-reference frequency $g_s$ at each position carrying a valid genotype, as follows: $g_s=0$ if the sample carried the reference allele, $g_s=$one if it carried the alternative allele, $g_s=0.5$ if both alleles were present. The distance between two samples at that position was then estimated by: $d = g_1(1\ g2) + g_2(1\ g1)$ where $g_1$ and $g_2$ are the $g_s$ values for the two samples. The pairwise distance was estimated as the mean of $d$ across all positions where $d$ could be computed (i.e. where neither of the two samples had a missing call). Neighbour-joining trees (NJTs) were then produced using the nj implementation in the R package ape (RRID:SCR_017343) on R v4.0.2 (RRID:SCR_001905, http://www.r-project.org/) from square distance matrices.

## Results

### Collaborations, site selection, and sample collections

As of August 2019, GenRe-Mekong has partnered with NMCPs in five countries to conduct large-scale genetic surveillance (Vietnam, Laos), smaller-scale pilot projects (Cambodia, Thailand), and epidemiological surveys (Bangladesh). GenRe-Mekong also worked with large-scale research projects investigating drug efficacy and malaria risk, or piloting elimination interventions. A total of 9623 samples from eight countries have been processed in this period (*Figure 1—figure supplement 1*). The majority of samples (n=6905, 72%) were collected in GMS countries (Vietnam, Laos, Cambodia, Thailand, Myanmar), but GenRe-Mekong also supported projects submitting samples from Bangladesh, India, and DR Congo (*Supplementary file 2*). The vast majority of processed samples were collected prospectively, under partnership agreements with GenRe-Mekong (n=9002, 93.5%); two research projects submitted retrospective samples collected in the period 2012–2015 (n=621, 6.5%, *Figure 1—figure supplement 1*). Approximately 59% of samples (n=5716) were submitted by NMCP partnerships, whose contribution increased over time as surveillance projects ramped up (43.4% in 2016, vs. 94.6% in 2018, *Figure 1—figure supplement 2*). Details of the partnerships, the nature of the studies conducted and the number of processed samples are given in *Table 1*.

Partnerships with NMCPs are often supported through collaborations with local malaria research groups, which provide support in implementing sample collections, and assist in the interpretation of results. To facilitate implementation in public health infrastructures, GenRe-Mekong provides template study protocols and associated documents; standardized kits of collection materials and documentation; and training for field and health centre staff. Study protocols are adapted to harmonize with local practices, and then approved by both a local ethical review board and the Oxford Tropical Research Ethics Committee (OxTREC). Informed consent forms and participant information sheets are translated to the local language(s), and public health facility staff are trained to execute sample collection procedures. Collection sites are mostly district-level or subdistrict-level health facilities, selected by NMCPs to cover the most informative endemic areas, often based on reported prevalence (*Figure 1*). Research studies and elimination projects included in their study protocol a sample collection procedure compatible with the standard GenRe-Mekong procedure, and sites were selected based on the study's requirements.

### Sample processing and genotyping

GenRe-Mekong samples consist of dried blood spots (DBSs) on filter paper. DNA extracted from the samples was selectively amplified (*Oyola et al., 2016*) to increase the proportion of parasite DNA and reduce human DNA contamination before genotyping (see Materials and methods). The production of genetic report cards involves genotyping different types of variants: single nucleotide polymorphisms (SNPs), copy number variations and sequences of gene domains. These operations

**Table 1.** Participating studies in GenRe-Mekong.

For each study, we list the NMCP and Research partners involved, the type of study, the geographical region covered and the number of collection sites. In the last two columns, we show the total number of samples submitted, and the number included in the final set of quality-filtered samples used in epidemiology analyses.

| NMCP partner | Research / technical partner | Study type | Regions surveyed | Sites | Submitted samples | Filtered samples |
|---|---|---|---|---|---|---|
| Center for Malaria Parasitology and Entomology of Lao PDR (CMPE) | Lao-Oxford-Mahosot Hospital-Wellcome Trust Research Unit (LOMWRU), Vientiane | Genetic Surveillance | South Laos (five provinces) | 51 | 1555 | 1387 |
| Institute of Malariology, Parasitology, and Entomology Quy Nhon (IMPE-QN), Vietnam | Oxford University Clinical Research Unit (OUCRU), Ho Chi Minh City | Genetic Surveillance | Central Vietnam (seven provinces) | 51 | 1632 | 1492 |
| National Institute of Malariology, Parasitology, and Entomology (NIMPE), Vietnam | Vysnova Partners, Mahidol-Oxford Research Unit (MORU) | Epidemiological Study | South Vietnam (three provinces) | 19 | 292 | 265 |
| National Center for Parasitology, Entomology, and Malaria Control (CNM), Cambodia | | Genetic Surveillance | Northeast Cambodia (two provinces) | 19 | 182 | 174 |
| Bangladesh National Malaria Control Programme | Mahidol-Oxford Research Unit (MORU) | Epidemiological Study | Bangladesh (Chittagong Division) | 55 | 2055 | 1575 |
| - | Mahidol-Oxford Research Unit (MORU) | Clinical Efficacy Study | Cambodia, Vietnam, Thailand, Lao PDR, Myanmar, Bangladesh, India, DR Congo | 17 | 1875 | 1123 |
| - | National Institutes of Health (NIH) | Clinical Efficacy Study | Cambodia | 3 | 592 | 502 |
| - | Oxford University Clinical Research Unit (OUCRU) | Epidemiological Study | South Vietnam | 4 | 184 | 175 |
| - | Mahidol-Oxford Research Unit (MORU) | Elimination Study | West Cambodia | 1 | 69 | 32 |
| - | Mahidol-Oxford Research Unit (MORU) | Epidemiological Study | Northeast Thailand | 7 | 87 | 60 |
| - | Shoklo Malaria Research Unit (SMRU) | Clinical Efficacy Study | Thailand (Tak province) | 4 | 29 | 28 |
| - | Shoklo Malaria Research Unit (SMRU) | Elimination Study | Myanmar (Kayin State) | 51 | 1071 | 813 |
| Total | | | | | 9623 | 7626 |

were performed by SpotMalaria, the genotyping platform underpinning GenRe-Mekong, whose implementation evolved during the course of the project; details of the methods used in different versions are provided in the Supplementary Materials. In the initial phase, SpotMalaria used a mixture of technologies: capillary sequencing of the *kelch13* gene to detect SNPs associated with artemisinin resistance (*Ashley et al., 2014*; *Ariey et al., 2014*); and high-throughput mass spectrometry to genotype SNP variants. This was later replaced with an amplicon sequencing process, based on short-read deep sequencing of specific portions of the parasite genome, supporting a high degree of multiplexing (see Materials and methods). A total of 3473 samples (36%) were processed by the amplicon sequencing platform, which delivered a higher genotyping success rate than the earlier process (94% vs 82% mean success rate for genetic barcode positions).

The vast majority of samples were taken from malaria patients upon admission (92%, n=8866). The remainder were from recurrent clinical episodes, or collected as part of post-admission time series to study infection dynamics (n=757, 7.9%), and were excluded from epidemiological analyses in order to minimize biases and avoid duplicates. Genotypes at mitochondrial positions provided confirmation of the infecting parasite species: *P. falciparum* (*Pf*), *P. vivax* (*Pv*), *P. knowlesi* (*Pk*), *P. malariae* (*Pm*), and *P. ovale* (*Po*). All five species were detected in our dataset: non-Pf parasites were found in 8.8% of samples (n=745 out of 8486 samples for which species could be determined). A proportion of samples (n=414, 4.9%) only tested positive for non-*Pf* species, possibly due to misdiagnosis or extremely low *Pf* parasitaemia, and were excluded from epidemiological analyses. *Pv* was

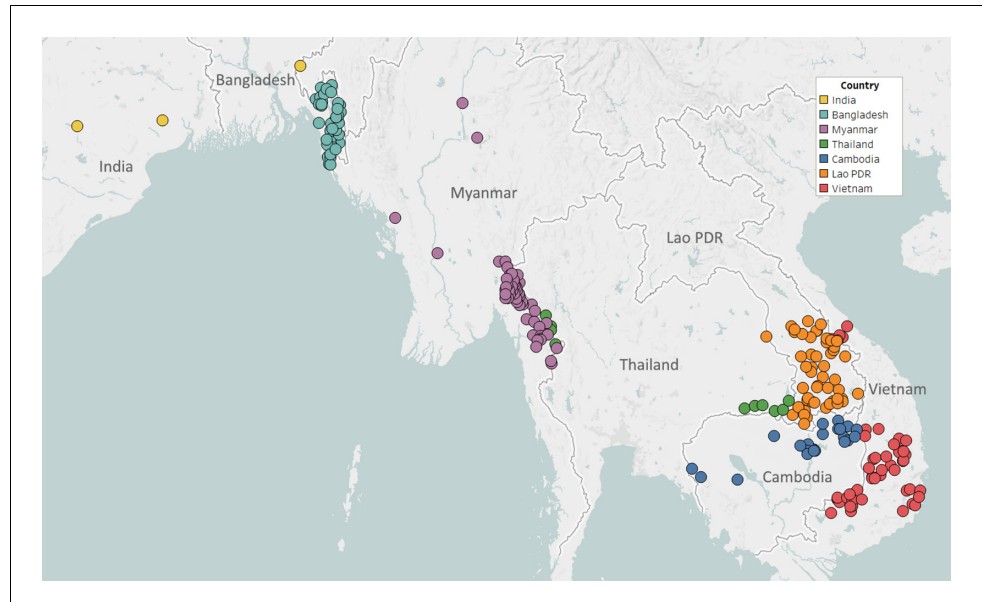

**Figure 1.** Map of GenRe-Mekong sample collection sites in Asia. Sites markers are colored by country. One site in Kinshasa (DR Congo) not shown.

The online version of this article includes the following figure supplement(s) for figure 1:

**Figure supplement 1.** Number of samples collected prospectively by month in each country.
**Figure supplement 2.** Trends in sample collections over time.

the most commonly detected non-*Pf* species (317 *Pf/Pv* mixed infections, and 405 *Pv*-only infections), followed by *Pk* (11 *Pf/Pk* and 6 *Pk*-only infections), while *Pm* and *Po* were detected in three and two samples, respectively.

## Genetic barcodes

GenRe-Mekong produces a *genetic barcode* for each sample to enable analyses of relatedness, diversity, multiplicity of infection and population structure. Genetic barcodes are constructed by concatenating the alleles at 101 SNPs distributed across all nuclear chromosomes (see Materials and methods), chosen on the basis of their geographically widespread variability and their power to recapitulate genetic distance. Genetic barcodes can be used to detect loss of diversity due to demographic effects, (*Daniels et al., 2015*) or to compare parasites from the same patient to distinguish recrudescences from reinfections (*Felger et al., 2020*). They can also produce estimates of genetic distance, which may not be sufficiently accurate for detailed inferences, but are useful for visualizing macroscopic population-level features. For example, a neighbor-joining tree derived from these genetic distance estimates (*Figure 2*) clearly separates parasites from the Thai-Myanmar border region from those circulating along the Thai-Cambodian border, consistent with findings from WGS analyses (*Miotto et al., 2015*). Hence, while genetic barcodes produce lower resolution results than WGS data, they could be used for rapid low-cost detection of candidate imported parasites, to be further analysed using higher-definition approaches. We used genetic barcode results to discard 827 samples that failed to produce barcodes due to low *Pf* DNA content. This yielded a final set of 7626 *Pf* samples, corresponding to 90.2% of all *Pf*-containing samples taken upon admission, which provided the data used for epidemiological analyses.

## Survey of drug resistance mutations

GenRe-Mekong produces genotypes covering a broad range of known variants associated to drug resistance (*Table 2*) to support assessment of the spread and risk of drug resistance. The interpretation of these genetic markers in phenotypic terms requires extensive knowledge of relevant literature, which is often outside the domain of expertise of public health officers. To bridge this gap, we use genotypes to derive *predicted phenotypes* based on a set of rules derived from peer-reviewed

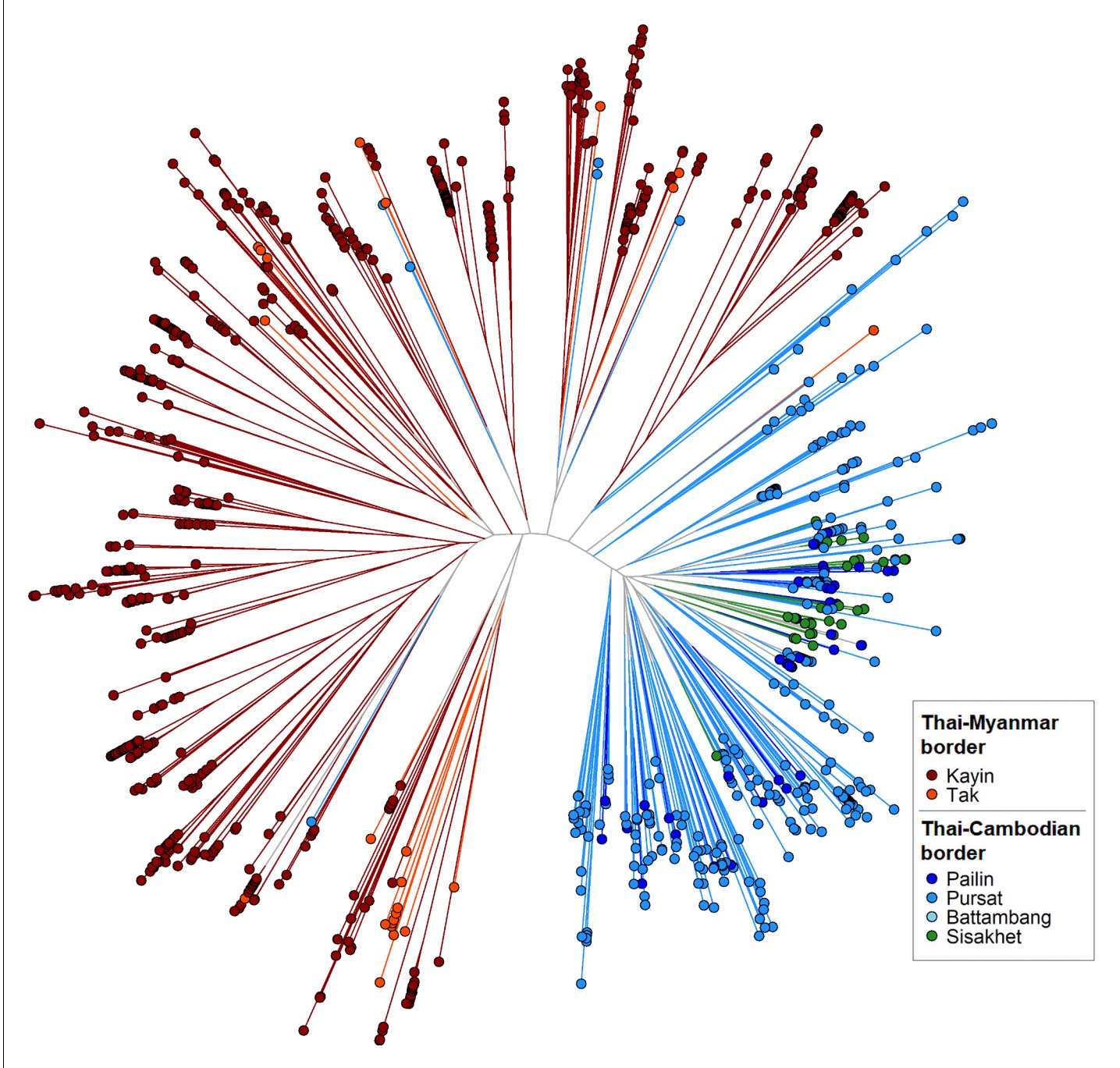

**Figure 2.** Neighbor-joining tree using barcode data to show genetic differentiation between parasites in the Thai-Myanmar and Thai-Cambodian border regions. The tree was derived from a matrix distance matrix, computed by comparing the genetic barcodes of samples. The branch length separating each pair of parasites represents the amount of genetic differentiation between them: individuals separated by shorter branches are more similar to each other. Samples from provinces/states of Myanmar, Thailand, and Cambodia near to the borders were included. Each circular marker represents a sample, colored by the province/state of origin.

publications (see Materials and methods and formal rules definitions available from the article's Resource Page). These rules predict samples as *resistant* or *sensitive* to a particular drug or treatment, or *undetermined*. Since our procedures do not include the measurement of clinical or in vitro phenotypes, we are only able to predict a drug resistant phenotype based on known associations of certain markers with resistance to certain drugs. Although we report a large catalogue of variations

**Table 2.** Drug resistance-related SNPs genotyped by GenRe-Mekong (excludes *kelch13*).

| Chromosome | Position | Gene Id | Gene Description | Mutation | Reference | Alternate |
|---|---|---|---|---|---|---|
| Pf3D7_04_v3 | 748239 | | *dhfr* (bifunctional dihydrofolate reductase-thymidylate synthase) | N51I | A | T |
| Pf3D7_04_v3 | 748262 | | | C59R/Y | T | C |
| Pf3D7_04_v3 | 748263 | PF3D7_0417200 | | C59R/Y | G | A |
| Pf3D7_04_v3 | 748410 | | | S108N/T | G | AC |
| Pf3D7_04_v3 | 748577 | | | I164L | A | T |
| Pf3D7_05_v3 | 958145 | | *mdr1* (multidrug resistance protein 1) | N86Y | A | T |
| Pf3D7_05_v3 | 958440 | PF3D7_052300 | | Y184F | A | T |
| Pf3D7_05_v3 | 961625 | | | D1246Y | G | T |
| Pf3D7_07_v3 | 403623 | | *crt* (chloroquine resistance transporter) | N75D/E | T | A |
| Pf3D7_07_v3 | 403625 | PF3D7_0709000 | | K76T | A | C |
| Pf3D7_07_v3 | 405362 | | | N326S | A | G |
| Pf3D7_07_v3 | 405600 | | | I356T | T | C |
| Pf3D7_08_v3 | 549681 | | *dhps* (dihydropteroate synthetase) | S436A/Y/F/G | T | GC |
| Pf3D7_08_v3 | 549682 | | | S436A/Y/F/G | C | TAG |
| Pf3D7_08_v3 | 549685 | PF3D7_0810800 | | A437G | G | C |
| Pf3D7_08_v3 | 549993 | | | K540E/N | A | GT |
| Pf3D7_08_v3 | 549995 | | | K540E/N | A | TG |
| Pf3D7_08_v3 | 550117 | | | A581G | C | G |
| Pf3D7_08_v3 | 550212 | | | A613S/T | G | TA |
| Pf3D7_13_v3 | 748395 | PF3D7_1318100 | *fd* (ferredoxin) | D193Y | C | A |
| Pf3D7_13_v3 | 2504560 | PF3D7_1362500 | *exo* (exonuclease) | E415G | A | G |
| Pf3D7_14_v3 | - | PF3D7_1408000 and PF3D7_1408100 | *pm23* (plasmepsin 2 and plasmepsin 3) | Breakpoint | - | - |
| Pf3D7_14_v3 | 1956225 | PF3D7_1447900 | *mdr2* (multidrug resistance protein 2) | T484I | G | A |
| Pf3D7_14_v3 | 2481070 | PF3D7_1460900 | *arps10* (apicoplast ribosomal protein S10) | V127M | G | A |
| Pf3D7_14_v3 | 2481073 | | | D128Y/H | G | TC |

which have been associated with resistance, we do not use all variations to predict resistance. Rather, our predictive rules are conservative and only use markers that have been strongly characterized and validated in published literature and shown to play a crucial role in clinical or in vitro resistance. These critical variants include single nucleotide polymorphisms (SNPs) in genes *kelch13* (resistance to artemisinin), (*Ariey et al., 2014*) *crt* (chloroquine), *dhfr* (pyrimethamine), *dhps* (sulfadoxine), as well as an amplification breakpoint sequence in *plasmepsin2/3* (marker of resistance to piperaquine) (*Amato et al., 2017*). In addition, we report several additional variants found in drug resistance backgrounds but not used to predict resistance, such as mutations in *mdr1* (linked to resistance to multiple drugs), components of the predisposing ART-R background *arps10*, *ferredoxin*, *mdr2* (*Miotto et al., 2015*), and the *exo* marker associated with resistance to piperaquine (*Amato et al., 2017*). Several samples had missing genotype calls which were required for phenotype prediction; therefore, we also devised a number of rules for *imputation* of missing genotypes based on information from linked alleles. These imputation rules (see Materials and methods) are based on an analysis of allele associations using data from over 7000 samples in the MalariaGEN *Pf* Community Project (*Pearson et al., 2019*) and are applied prior to phenotype prediction rules. Phenotypic predictions allow simple estimations of the proportions of resistant parasites at the population level, which can be readily tabulated and mapped for use in public health decision-making. By aggregating sample data at various geographic levels (site, district, province, region, country),

GenRe-Mekong delivers to NMCPs maps that capture the current drug resistance landscape, and can be compared to detect changes over time. Most GenRe-Mekong maps use intuitive 'traffic light' color schemes, in which red signifies presence of resistance, and green its absence. Below, we illustrate some results at regional level for the GMS and nearby countries, which are also summarized in *Table 3*.

The spread of artemisinin resistance (ART-R) is an urgent concern in the GMS. We estimated frequencies of predicted ART-R parasites based on the presence of nonsynonymous mutations in the *kelch13* gene, as listed by the **World Health Organization, 2018**. The resulting map indicates that ART-R has reached very high levels in the lower Mekong region (Cambodia, northeastern Thailand, southern Laos, and Vietnam), nearing fixation in Cambodia and around its borders, with the exception of very few provinces of Laos and the Vietnam coast (*Figure 3A*). Predicted ART-R frequencies decline to the west of this region: no samples in this study were predicted to be ART-R in India and Bangladesh, thus showing no evidence of spread beyond the GMS, or of local emergence of resistant parasite populations. An analysis of the distribution of *kelch13* ART-R alleles (*Figure 3—figure supplement 1*, *Supplementary file 3*) reveals a marked difference between the lower Mekong

**Table 3.** Frequencies of resistant parasites in provinces/states/divisions surveyed, for different antimalarials.

| Country | Province, State, or Division | ART-R | PPQ-R | DHA-PPQ-R | CQ-R | PYR-R | SD-R | SP-R | SP-R (IPTp) |
|---|---|---|---|---|---|---|---|---|---|
| India | Odisha | 0% | 0% | 0% | 18% | 57% | 6% | 1% | 0% |
| | West Bengal | 0% | 0% | 0% | 47% | 71% | 14% | 5% | 0% |
| | Tripura | 0% | 0% | 0% | 85% | 100% | 99% | 55% | 0% |
| Bangladesh | Chittagong | 0% | 0% | 0% | 97% | 100% | 87% | 46% | 16% |
| Myanmar | Rakhine | 0% | 0% | 0% | 71% | 100% | 100% | 51% | 26% |
| | Bago | 1% | 0% | 0% | 88% | 100% | 100% | 91% | 74% |
| | Mandalay | 29% | 0% | 0% | 96% | 98% | 98% | 29% | 24% |
| | Kayin | 54% | 2% | 0% | 100% | 100% | 56% | 73% | 27% |
| Thailand | Tak | 61% | - | 0% | 100% | 100% | 96% | 100% | 88% |
| | Sisakhet | 100% | 90% | 90% | 100% | 100% | 100% | 100% | 100% |
| | Ubon Ratchathani | 80% | 75% | 56% | 100% | 100% | 85% | 100% | 17% |
| Cambodia | Pailin | 93% | 97% | 90% | 100% | 100% | 100% | 100% | 56% |
| | Battambang | 100% | 100% | 100% | 100% | 100% | 88% | 100% | 29% |
| | Pursat | 88% | 98% | 67% | 100% | 100% | 92% | 98% | 44% |
| | Preah Vihear | 61% | 100% | 11% | 100% | 100% | 94% | 98% | 21% |
| | Steung Treng | 93% | 75% | 70% | 100% | 100% | 97% | 100% | 0% |
| | Ratanakiri | 49% | 79% | 42% | 99% | 100% | 76% | 90% | 5% |
| Laos | Champasak | 66% | 75% | 56% | 100% | 100% | 88% | 94% | 12% |
| | Attapeu | 46% | 43% | 31% | 100% | 100% | 82% | 100% | 18% |
| | Sekong | 26% | 6% | 0% | 100% | 100% | 91% | 74% | 5% |
| | Salavan | 17% | 2% | 1% | 89% | 97% | 28% | 38% | 1% |
| | Savannakhet | 10% | 1% | 0% | 87% | 96% | 21% | 41% | 2% |
| Vietnam | Binh Phuoc | 92% | 93% | 83% | 100% | 100% | 100% | 100% | 14% |
| | Dak Nong | 94% | 92% | 88% | 100% | 100% | 97% | 96% | 22% |
| | Dak Lak | 96% | 90% | 86% | 100% | 100% | 100% | 99% | 15% |
| | Gia Lai | 84% | 83% | 76% | 99% | 100% | 98% | 95% | 4% |
| | Khanh Hoa | 22% | 5% | 2% | 95% | 100% | 97% | 74% | 2% |
| | Ninh Thuan | 13% | 18% | 0% | 28% | 100% | 98% | 75% | 0% |
| | Quang Tri | 16% | 9% | 0% | 75% | 76% | 59% | 26% | 5% |
| Congo PDR | Kinshasa | 0% | 0% | 0% | 58% | 98% | 72% | 88% | 0% |

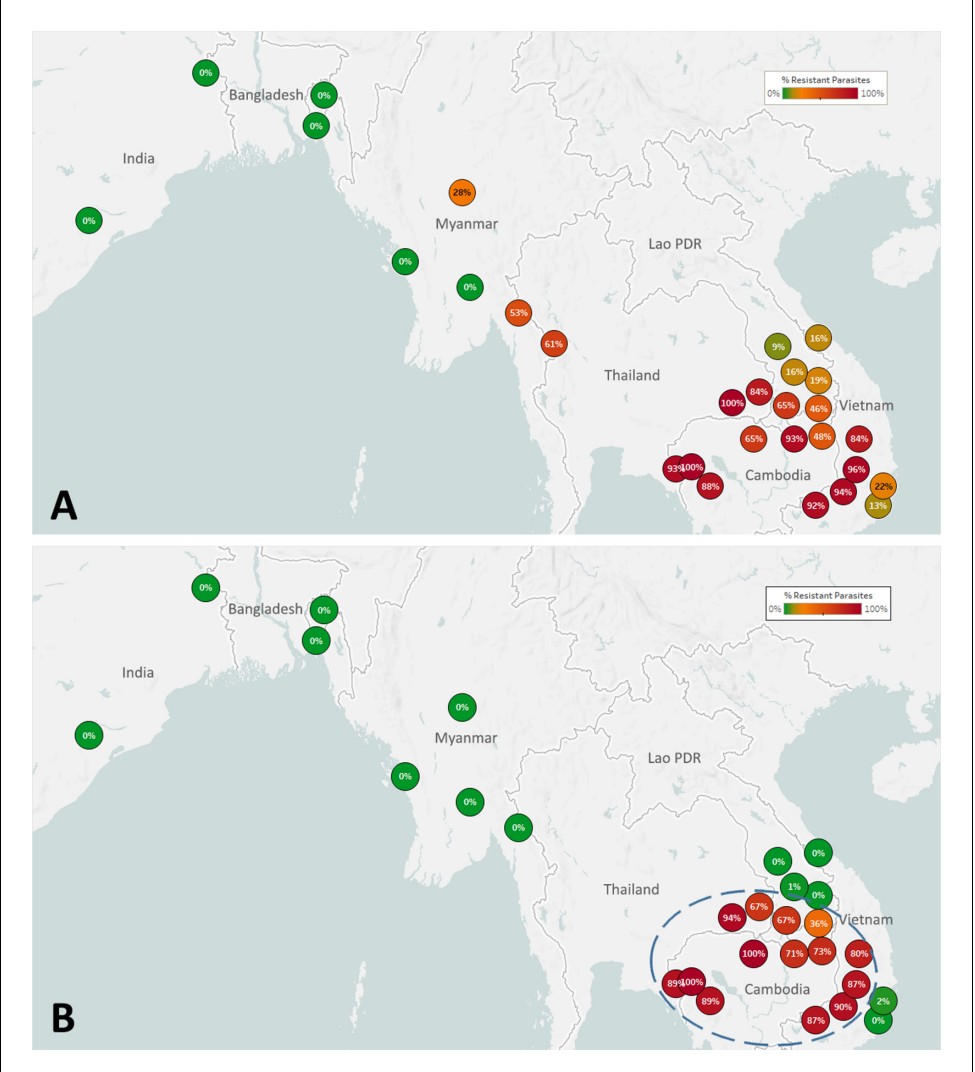

**Figure 3.** Map of the spread of (**A**) artemisinin resistance (ART-R) and (**B**) dihydroartemisinin-piperaquine resistance (DHA-PPQ-R) in Asian countries. Marker text and color indicate the proportion of sample classified as resistant in each province/state/division surveyed. A total of 6762 samples were included in (**A**) and 3395 samples in (**B**), after excluding samples with undetermined phenotype prediction. The results are summarized in *Table 3*. The online version of this article includes the following source data and figure supplement(s) for figure 3:

**Source data 1.** Proportions of parasites predicted to be resistant to artemisinin and to the DHA-PPQ combination therapy in each province/state/division.

**Figure supplement 1.** *kelch13* allele diversity in Asian countries.

**Figure supplement 1—source data 1.** Sample frequencies for different kelch13 alleles at each province/state/division.

**Figure supplement 2.** Map of Piperaquine Resistance (PPQ-R) in Asian countries.

**Figure supplement 2—source data 1.** Proportions of parasites predicted to be resistant to piperaquine in each province/state/division.

**Figure supplement 3.** Map of Chloroquine Resistance (CQ-R) in Asian countries.

**Figure supplement 3—source data 1.** Proportions of parasites predicted to be resistant to chloroquine in each province/state/division.

**Figure supplement 4.** Map of Pyrimethamine Resistance (PYR-R) in Asian countries.

**Figure supplement 4—source data 1.** Proportions of parasites predicted to be resistant to pyrimethamine in each province/state/division.

**Figure supplement 5.** Map of Sulfadoxine Resistance (SD-R) in Asian countries.

**Figure supplement 5—source data 1.** Proportions of parasites predicted to be resistant to sulfadoxine in each province/state/division.

region, where the *kelch13* C580Y mutation is the dominant allele, and the region comprising Myanmar and western Thailand, where a wide variety of non-synonymous *kelch13* variants are found, and C580Y is not dominant. This reflects a recent increase of C580Y mutant prevalence in Cambodia and neighboring regions, resulting from the rapid spread of the KEL1/PLA1 strain of multidrug-resistant parasites (*Hamilton et al., 2019*; *Amato et al., 2018*). This hard selection sweep has replaced a variety of ART-R alleles previously present in that region, resulting from multiple soft sweeps (*Miotto et al., 2015*; *Miotto et al., 2013*); this process has not occurred along the Thai-Myanmar border, where allele diversity is still very pronounced. The spread of DHA-PPQ resistant (DHA-PPQ-R) strains in the lower Mekong region is confirmed when we map the frequency of *plasmepsin2/3* amplifications conferring piperaquine resistance (PPQ-R, *Figure 3—figure supplement 2*), which occur where C580Y is most prevalent. Mapping the combined presence of C580Y and *plasmepsin2/3* amplification shows that parasites carrying both markers are confined to a well-defined area of the lower Mekong region, and these resistant strains have not made their way into provinces of Laos and Vietnam where ART-R and PPQ-R alleles circulate separately (*Figure 3B*). Over time, GenRe-Mekong will continue to track across the region the spread of strains carrying drug resistance mutations.

Resistant populations can revert to sensitive haplotypes after drugs are discontinued, as was the case for chloroquine-resistant parasites in East Africa (*Laufer et al., 2006*; *Frosch et al., 2014*). To help detect similar trends in the GMS, GenRe-Mekong reports on markers of resistance to previous frontline antimalarials that have been discontinued because of reduced efficacy. The resulting data show that, decades after the replacement of chloroquine as frontline therapy, the frequency of parasites predicted to be resistant (CQ-R) remains exceptionally high across the GMS (*Figure 3—figure supplement 3*). The reasons for such sustained levels of resistance are unclear; the continued use of chloroquine as frontline treatment for *P. vivax* malaria, and the low diversity associated with the extremely high prevalence of resistant haplotypes could be major contributing factors. Similarly, we found high levels of the *dhfr* and *dhps* markers associated with resistance to sulfadoxine-pyrimethamine (SP, *Figure 3—figure supplements 4* and *5*). It is unclear why resistance to SP is so widespread, several years after discontinuing this therapy in the GMS, although similar results have been seen in Malawi (*Artimovich et al., 2015*). Again, very low haplotype diversity may be an obstacle to reversion, and it is also possible that compensatory changes have minimized the fitness impact of resistant mutations over time, diminishing the pressure to revert. It is interesting that predicted resistance is lowest in India, where SP is still used with artesunate as the frontline ACT (*Directorate of National Vector Borne Disease Control Programme DGoHS and Government of India, 2013*).

## Case study: Vietnam

In Vietnam, sample collections were carried out by two NMCP institutes (IMPE-QN and NIMPE), covering approximately 70 sites in seven provinces. Genetic report cards were delivered to public health officials over two malaria seasons (*Figure 4*), communicating new findings for malaria control. Prior to this surveillance activity, evidence of artemisinin resistance had been found in the provinces of Binh Phuoc, Gia Lai, Dak Nong, Khanh Hoa, and Ninh Thuan province (*World Health Organization, 2017*). GenRe-Mekong data confirmed the presence of parasites carrying ART-R markers in these provinces, and showed that the province of Dak Lak also has extremely high levels of predicted ART-R (*Figure 4—figure supplement 1*). Furthermore, our data showed that nearly all ART-R parasites collected near the border with were also predicted to be PPQ-R, in that they carried both the *kelch13* C580Y mutation (*Figure 4—figure supplement 2*) and *plasmepsin2/3* amplification (*Hamilton et al., 2019*; *Amato et al., 2018*). C580Y parasites were also found in the coastal provinces of Ninh Thuan, Khanh Hoa and Quang Tri, but they did not carry the PPQ-R marker; it is therefore likely the *kelch13* mutations were introduced by an earlier sweep of ART-R parasites. Several parasites in Khang Hoa carried the *kelch13* P553L mutation, previously associated with an ART-R founder population in Binh Phuoc province (*Miotto et al., 2015*; *Takala-Harrison et al., 2015*), supporting the hypothesis they belong to an earlier sweep (*Figure 4—figure supplement 2*).

Data from consecutive seasons offers a view of the dynamics of drug resistance spread. In the 2018/2019 season, there was a marked increase in the number of cases in the Krong Pa district of Gia Lai province (*Figure 4*). In 2017/2018, this district accounted for 15% of cases in the three central provinces that border with Cambodia (n=96 of 656); the following season, this increased to 64% (n=341 of 529, $p < 10^{-15}$). In the same timeframe, predicted DHA-PPQ-R parasites in Krong Pa rose

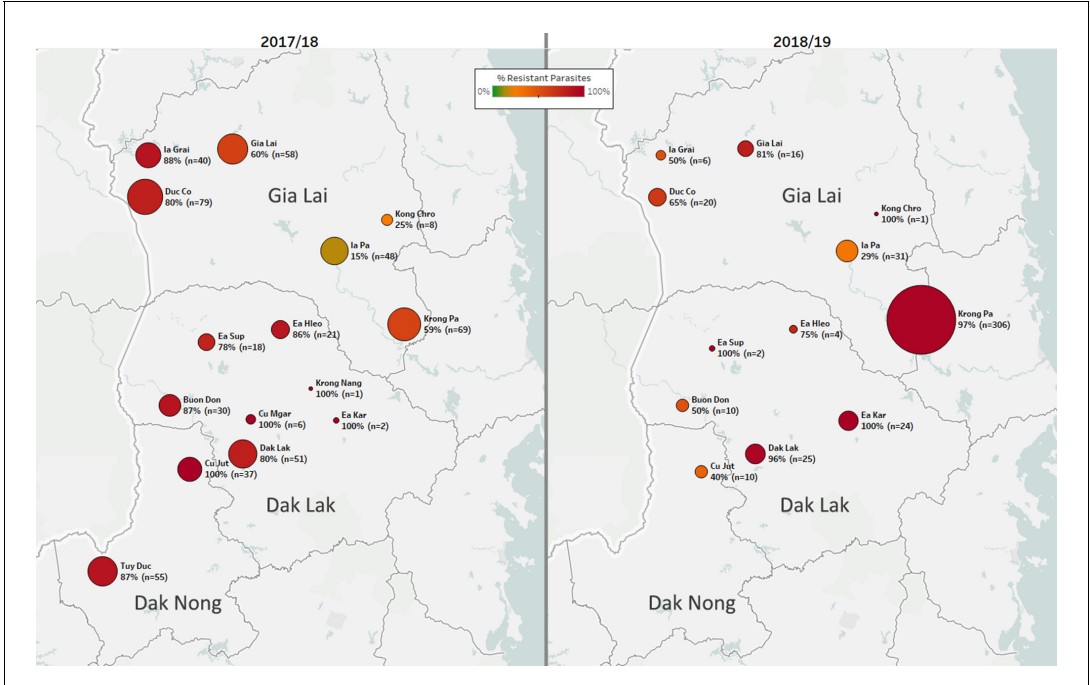

**Figure 4.** Longitudinal sample counts and proportions of DHA-PPQ-R parasites in three provinces of Central Vietnam. The same geographical area (Gia Lai, Dak Lak, and Dak Nong provinces) is shown for two malaria seasons: 2017/18 (12 months from May 2017, n=523) and 2018/2019 (the following 12 months, n=455). Districts are represented by markers whose size is proportional to the number of samples, and whose color indicates the frequency of samples carrying both the *kelch13* C580Y mutation and the *plasmepsin2/3* amplification, and thus predicted to be DHA-PPQ-R. Marker labels show district name, resistant parasite frequency, and sample count.

The online version of this article includes the following source data and figure supplement(s) for figure 4:

**Source data 1.** Proportions of samples predicted to be resistant to DHA-PPQ in districts of Vietnam, in the seasons 2017/18 and 2018/19.

**Figure supplement 1.** Frequencies of ART-R and PPQ-R parasites in Vietnam.

**Figure supplement 1—source data 1.** Counts and proportions of samples predicted to be resistant to artemisinin, piperaquine and DHA-PPQ in provinces of Vietnam.

**Figure supplement 2.** Distribution of kelch13 alleles in seven provinces of Vietnam.

**Figure supplement 2—source data 1.** Sample frequencies for different *kelch13* alleles in provinces of Vietnam.

from 65% (n=40 of 62) to 98% (n=298 of 305, p<10$^{-14}$). These results suggest that an outbreak occurred in this district in 2018/2019, underpinned by strong selection of a genetic background able to survive the frontline ACT DHA-PPQ.

## Case study: Laos

The Lao NMCP implemented genetic surveillance in five provinces of southern Laos, at over 50 public health facilities. Artemisinin-resistant parasites were found in all five provinces, at frequencies higher in districts bordering Thailand and Cambodia (*Figure 5A*). The *kelch13* C580Y mutation was found in four of the five provinces, and was the most common ART-R allele (*Figure 5—figure supplement 1*). However, parasites carrying both C580Y and the *plasmepsin2–3* amplification were restricted to the two southernmost provinces (Champasak and Attapeu, referred to as 'Lower Zone', *Figure 5B*), and completely absent from Savannakhet and Salavan provinces ('Upper Zone') where C580Y parasites lack the PPQ-R amplification. In other words, it appears that DHA-PPQ-R parasites, possibly imported from Cambodia or Thailand, have migrated into the Lower Zone but not the Upper Zone, where a different population of ART-R parasites circulates.

Given the very recent aggressive spread of DHA-PPQ-R strains, it is likely that ART-R parasites in the Upper Zone are remnants of an earlier sweep which may also have spread from the south, as suggested by the higher frequency in Salavan province than in Savannakhet. To confirm the presence of distinct ART-R populations, we used genetic barcodes to construct a tree that recapitulates

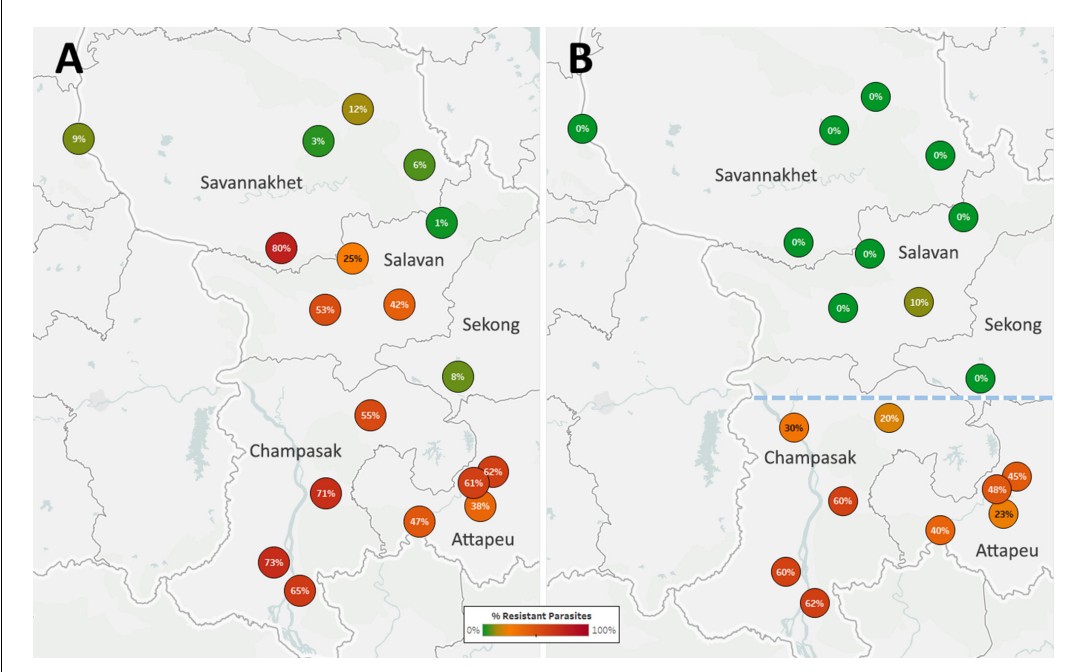

**Figure 5.** Proportions of ART-R and KEL1/PLA1 parasites in southern Laos districts. Districts in five provinces of southern Laos are represented by markers whose color and label indicates the frequency of samples classified as ART-R (**A**) and as DHA-PPQ-R, i.e. possessing markers of resistance to both artemisinin and piperaquine (**B**). Only districts with more than 10 samples with valid genotypes are shown. In panel (**B**), a dashed line denotes a hypothetical demarcation line between a Lower Zone, where DHA-PPQ-R strains have spread, and an Upper Zone, where they are absent and ART-R parasites belong to different strains.

The online version of this article includes the following source data and figure supplement(s) for figure 5:

**Source data 1.** Counts and proportions of samples predicted to be resistant to artemisinin and DHA-PPQ in districts of Laos.
**Figure supplement 1.** Frequencies Distribution of *kelch13* alleles in five provinces of Laos.
**Figure supplement 1—source data 1.** Sample frequencies for different kelch13 alleles in provinces of Laos.
**Figure supplement 2.** Neighbour-joining tree using barcode data to show genetic differentiation between groups of parasites collected in Southern Laos.

population structure in Laos (*Figure 5—figure supplement 2*), which clearly separates Upper Zone and Lower Zone parasites. In this tree, DHA-PPQ-R parasites form a large, tight cluster clearly separated from the *kelch13* wild-type samples from the Upper Zone. The Upper Zone C580Y mutants cluster separately from both these groups, and appear more similar to some C580Y mutants from the Lower Zone which do not carry the PPQ-R amplification, corroborating the hypothesis that Upper Zone mutants migrated from the South. It is likely that the northward spread of DHA-PPQ-R strains has been contained by the use of artemether-lumefantrine in Laos, which diminishes the survival advantage of resistance to piperaquine. However, the spread of DHA-PPQ-R parasites across the Lower Zone, probably displacing previous ART-R strains, suggests that they are well-adapted and highly competitive even in the absence of pressure from piperaquine.

## Release of genetic report card data

GenRe-Mekong's primary data outputs are Genetic Report Cards, delivered as spreadsheets comprising sample metadata (time and place of collection), drug resistance genotypes and phenotype predictions, detected species and genetic barcodes. As soon as sample processing is complete, GRCs are returned to the stakeholders of the studies that contributed the samples, which typically include the NMCP and local scientific partners. Detailed analyses of GRC data may also be conducted by the GenRe-Mekong analysis team and local partners, and their results reported to the NMCP. On a regular basis, GRC data from all studies will be aggregated and released to public access, to benefit the research and public health community. The public releases are detailed by

sample, and comprise all genetic data and their derivatives such as phenotype predictions. The first public release is currently available from the article's Resource Page at https://www.malariagen.net/resource/29.

## Discussion

GenRe-Mekong provides a genetic surveillance platform suitable for endemic regions of low- and middle-income countries, which delivers to NMCPs detailed knowledge about the genetic epidemiology of malaria parasites, to support decision-making. Pilot studies have been conducted in all GMS countries, with the Vietnam and Laos NMCPs having implemented GenRe-Mekong on a long-term basis. GenRe-Mekong has multiple features that facilitate NMCP engagement: a sample collection procedure that easily integrates with standard medical facility workflows; standardized protocols and training to support implementation; clear presentation of results, including translation to phenotype predictions, to provide intuitive understanding and rapid communication; and support by our regional analysis team and local partners to deliver and discuss findings. GenRe-Mekong has also worked closely with research projects, contributing to their analyses of the genotyping data and supporting publication of key findings. The genetic data produced were valuable for a wide range of research applications, such as clinical studies of drug efficacy (*van der Pluijm et al., 2019*), evaluation of elimination interventions (*Landier et al., 2018*), and epidemiological investigation of malaria importation (*Chang et al., 2019*).

Collaborations with public health organizations have rapidly translated into real impact for malaria control, especially where GenRe-Mekong has been implemented over multiple seasons. Genetic surveillance results were used by the Vietnam NMCP and Ministry of Health in reviews of national drug policy, leading to the replacement of DHA-PPQ with artesunate-pyronaridine as frontline therapy in four provinces. These included the province of Dak Lak, where an early report by GenRe-Mekong in 2018 was the first evidence of ART-R, confirmed by treatment failure data from in vivo therapy efficacy studies (TES) in 2019. In addition, our report of a DHA-PPQ-R outbreak in Gia Lai province has alerted authorities to the need to review the use of DHA-PPQ in that province. In Laos, authorities have been equally responsive, using GenRe-Mekong reports in their review of frontline therapy choices: the Ministry of Health opted against adopting the DHA-PPQ ACT based on our evidence of the expansion of resistant strains in the Lower Zone of southern Laos. The impact has not been limited to the national level: data shared by surveillance and research projects participating in GenRe-Mekong has powered regional large-scale epidemiological analyses in the GMS and beyond, revealing patterns of spread and evolution of multidrug-resistant malaria (*van der Pluijm et al., 2019*). By combining results from areas populated by multidrug resistant strains with those from countries where these strains could potentially spread, such as Bangladesh and India, GenRe-Mekong maps support risk assessment and preparedness. GenRe-Mekong will continue to encourage public data sharing to increase the value of genetic data generated, while respecting patient anonymity and giving recognition to those who contributed to the project.

A major advantage of genetic surveillance, compared to more costly clinical studies, is the potential for dense coverage across all endemic areas, which can identify important spatial heterogeneities across the territory. For example, DHA-PPQ was adopted as frontline therapy in Thailand based on the drug's efficacy in the western provinces; genetic data about the rise in prevalence of DHA-PPQ-R strains in the northeast of the country would probably have led to a different recommendation, had that information been available. Similarly, our data suggests that a single efficacy study in Savannakhet province could have convinced authorities that DHA-PPQ was suitable for Laos, with potential disastrous effects in the southernmost provinces. The extensive coverage provided by GenRe-Mekong routine surveillance allowed a more balanced evaluation of resistant strains prevalence across all endemic provinces. In addition to dense coverage, genetic surveillance should also feature systematic and continued sampling over time, to support the detection of epidemiological changes, and also to allow prevalence comparisons between region, which is most meaningful when collection periods are matched.

The SpotMalaria genotyping platform is designed for extensibility, and has been expanded twice in the course of the project: to test for the newly discovered marker for the *plasmepsin2/3* amplification (*Amato et al., 2017*) and to add new mutations in *crt* which are associated to higher levels of piperaquine resistance in KEL1/PLA1 parasite (*Hamilton et al., 2019*; *Ross et al., 2018*;

*Agrawal et al., 2017*). Such improvements will continue as new markers are identified, and new techniques developed. However, there are newer drugs such as pyronaridine, and established drugs such as lumefantrine and amodiaquine, for which clinical drug resistance markers are yet to be identified. GenRe-Mekong will support the identification of new markers in practical ways, by performing WGS on selected surveillance samples, and contributing these data to public repositories to study epidemiological effects, such as reductions in diversity, increases in cases and founder populations, (*Miotto et al., 2013*) and to identify genomic regions under selection that may lead to discovering new markers. As the project develops, Genetic Report Cards will be expanded, to address new public health use cases, including those not directly related to drug resistance. For example, genetic barcodes and WGS data can be used to detect imported cases; to distinguish recrudescences from reinfections; and to measure connectedness between sites, and routes of spread (*Chang et al., 2019*).

GenRe-Mekong was conceived as a versatile and extensible platform that can be easily integrated in a wide range of endemic settings, at relatively low cost to allow extensive geographical coverage. These properties demand trade-offs, imposing certain limitations on the platform. First, we work with small-volume DBS samples, which makes sample collection easy to integrate in routine public health operations; however, low blood volumes mean low genotyping success rates from sub-microscopic infection, and thus GenRe-Mekong only processes samples from cases confirmed by microscopy or rapid diagnostic test (RDT). Second, we focus on genotypes that can be obtained from our high-throughput amplicon sequencing platform, allowing us to contain costs and manpower requirements. In some cases, we have to relax this restriction: for example, *mdr1* copy numbers currently cannot be reliably estimated from amplicon sequencing, because of the requirement for selective DNA amplification. Because of the importance of this genotype, we currently use an additional qPCR assay, but it is desirable to find innovative solutions that keep laboratory processes streamlined. Third, while our genetic barcodes can support useful analyses of populations, we plan to improve their resolution by including amplicons containing multiple highly polymorphic SNPs, which may be more informative of identity by descent.

In the future, the integration of genetic surveillance data in public health decision-making processes will be a major focus for GenRe-Mekong, to be addressed in several ways. First, we will make available online platforms for selecting, visualizing and retrieving genetic epidemiology data, which will provide customized views of the data. Second, we will integrate with public health information systems, such as NMCPs' dashboards, at both national and international level. This includes sharing GenRe-Mekong data through the World Health Organization's data visualization platform, Malaria Threats Map (http://apps.who.int/malaria/maps/threats/). Third, we will provide training and support to expand in-country expertise, developing local capacity to evaluate drug resistance data and other outputs that GenRe-Mekong will deliver in the future. Finally, we will promote in-country implementations of the SpotMalaria amplicon sequencing platform that underpins the system, to enable faster turnaround times and long-term self-sufficiency. As the adoption cycle continues, we envisage that a growing global network of public health experts will leverage on genetic surveillance to maximize the impact of their interventions, and accelerate progress toward malaria elimination.

## Acknowledgements

We are grateful all patients and health workers who participated in sample collections. This study used data from the MalariaGEN Pf3k Project and *Plasmodium falciparum* Community Project. We thank the staff of Wellcome Sanger Institute Sample Logistics, Sequencing, and Informatics facilities for their contribution; in particular, we are grateful to the Wellcome Sanger Institute DNA Pipelines Informatics team for supporting the development of the methods used in this work. We thank the many collaborators who contributed to the GenRe-Mekong Project, and especially: Pannapat Masingboon, Narisa Thongmee, Zoë Doran, Salwaluk Panapipat, Ipsita Sinha, Rapeephan Maude, Vilasinee Yuwaree, Tran Minh Nhat, Hoang Hai Phuc, Ro Mah Huan, Nguyen Minh Nhat, Tran Van Don. PR is a staff member of the World Health Organization; PR alone is responsible for the views expressed in this publication and they do not necessarily represent the decisions, policy or views of the World Health Organization. This work was supported, in whole or in part, by the Bill and Melinda Gates Foundation [OPP11188166, OPP1204268]. Under the grant conditions of the Foundation, a

## Additional information

### Competing interests

Hoa Nguyen, Nicole Zdrojewski, Sara Canavati: is an employee of Vysnova Partners Inc. The other authors declare that no competing interests exist.

### Funding

| Funder | Grant reference number | Author |
|---|---|---|
| Bill and Melinda Gates Foundation | OPP11188166 | Dominic P Kwiatkowski Olivo Miotto |
| Bill and Melinda Gates Foundation | OPP1204268 | Olivo Miotto |
| Wellcome Trust | 098051 | Dominic P Kwiatkowski |
| Wellcome Trust | 206194 | Dominic P Kwiatkowski |
| Wellcome Trust | 203141 | Dominic P Kwiatkowski |
| Wellcome Trust | 090770 | Dominic P Kwiatkowski |
| Wellcome Trust | 204911 | Dominic P Kwiatkowski |
| Wellcome Trust | 106698/B/14/Z | Dominic P Kwiatkowski |
| Medical Research Council | G0600718 | Dominic P Kwiatkowski |
| Department for International Development, UK Government | 201900 | Arjen M Dondorp |
| Department for International Development, UK Government | M006212 | Arjen M Dondorp |

The funders had no role in study design, data collection, data analysis, data interpretation, or report writing. The corresponding author had full access to all the data in the study and had final responsibility for the decision to submit for publication

### Author contributions

Christopher G Jacob, Resources, Data curation, Software, Formal analysis, Validation, Investigation, Methodology, Writing - original draft, Writing - review and editing; Nguyen Thuy-Nhien, Rob van der Pluijm, Sonexay Phalivong, Supervision, Investigation, Project administration; Mayfong Mayxay, Conceptualization, Resources, Data curation, Software, Formal analysis, Supervision, Funding acquisition, Investigation, Visualization, Methodology, Writing - original draft, Writing - review and editing; Richard J Maude, Huynh Hong Quang, Shavanthi Rajatileka, Anna E Jeffreys, Mehul Dhorda, Supervision, Investigation, Methodology, Project administration; Bouasy Hongvanthong, Viengxay Vanisaveth, Thang Ngo Duc, Huy Rekol, Rithea Leang, Xin Hui S Chan, Investigation, Project administration; Lorenz von Seidlein, Rick Fairhurst, François Nosten, Md Amir Hossain, Keobouphaphone Chindavongsa, Paul Newton, Elizabeth Ashley, Nicole Zdrojewski, Caterina I Fanello, Gilles Delmas, Dysoley Lek, Frank Smithuis, Tin Maung Hlaing, Parthasarathi Satpathi, Supervision, Investigation; Naomi Park, Pascal Ringwald, Investigation, Methodology; Scott Goodwin, Methodology; Rapeephan Maude, Cheah Huch, Le Thanh Dong, Kim-Tuyen Nguyen, Tran Minh Nhat, Tran Tinh Hien, Hoa Nguyen, Sara Canavati, Abdullah Abu Sayeed, Didar Uddin, Caroline Buckee, Marie Onyamboko, Thomas Peto, Rupam Tripura, Chanaki Amaratunga, Aung Myint Thu, Jordi Landier, Nguyen Hoang Chau, Seila Suon, James Callery, Podjanee Jittamala, Borimas Hanboonkunupakarn, Sasithon Pukrittayakamee, Aung Pyae Phyo, Khin Lin, Myo Thant, Sanghamitra Satpathi, Prativa K Behera, Amar Tripura, Subrata Baidya, Neena Valecha, Anupkumar R Anvikar, Akhter Ul Islam, Abul Faiz, Chanon Kunasol, Eleanor Drury, Mihir Kekre, Mozam Ali, Katie Love, Kate Rowlands, Christina S Hubbart,

Investigation; Daniel M Parker, Data curation, Investigation; Ranitha Vongpromek, Data curation, Supervision, Validation, Investigation, Methodology, Project administration; Namfon Kotanan, Data curation, Validation, Investigation, Methodology; Phrutsamon Wongnak, Data curation, Formal analysis, Investigation, Methodology; Jacob Almagro Garcia, Data curation, Software, Formal analysis, Investigation, Methodology; Richard D Pearson, Data curation, Software, Formal analysis, Methodology; Cristina V Ariani, Data curation, Formal analysis; Thanat Chookajorn, Data curation, Supervision, Project administration; Cinzia Malangone, Software, Supervision, Project administration; T Nguyen, Software; Jim Stalker, Software, Supervision; Ben Jeffery, Jonathan Keatley, Software, Supervision, Visualization; Kimberly J Johnson, Software, Visualization, Project administration; Dawn Muddyman, Software, Project administration; John Sillitoe, Resources, Supervision, Investigation, Project administration; Roberto Amato, Resources, Software, Supervision, Validation, Investigation, Project administration; Victoria Simpson, Conceptualization, Resources, Software, Supervision, Validation, Investigation, Project administration; Sonia Gonçalves, Conceptualization, Software, Supervision, Validation, Investigation, Methodology, Project administration; Kirk Rockett, Conceptualization, Formal analysis, Supervision, Validation, Investigation, Methodology, Project administration; Nicholas P Day, Resources, Formal analysis, Supervision, Validation, Investigation, Methodology, Project administration; Arjen M Dondorp, Resources, Formal analysis, Supervision, Investigation, Methodology, Project administration; Dominic P Kwiatkowski, Conceptualization, Resources, Supervision, Funding acquisition, Investigation, Project administration, Writing - review and editing; Olivo Miotto, Conceptualization, Resources, Data curation, Software, Formal analysis, Supervision, Funding acquisition, Investigation, Visualization, Methodology, Writing - original draft, Project administration, Writing - review and editing

### Author ORCIDs

Richard J Maude https://orcid.org/0000-0002-5355-0562
François Nosten http://orcid.org/0000-0002-7951-0745
Tran Minh Nhat https://orcid.org/0000-0002-9500-8341
Caroline Buckee https://orcid.org/0000-0002-8386-5899
Jordi Landier https://orcid.org/0000-0001-8619-9775
James Callery https://orcid.org/0000-0002-3218-2166
Frank Smithuis https://orcid.org/0000-0002-4704-9915
Christina S Hubbart https://orcid.org/0000-0001-9576-9581
Richard D Pearson https://orcid.org/0000-0002-7386-3566
Kirk Rockett https://orcid.org/0000-0002-6369-9299
Nicholas P Day https://orcid.org/0000-0003-2309-1171
Arjen M Dondorp https://orcid.org/0000-0001-5190-2395
Olivo Miotto https://orcid.org/0000-0001-8060-6771

### Ethics

Human subjects: For each country where the surveillance project was implemented in collaboration with public health authorities, we submitted a common GenRe-Mekong protocol, and obtained approval by a relevant local ethics review board and by the Oxford University Tropical Research Ethics Committee (OxTREC). In all countries we obtain informed consent from each malaria patient providing a sample, except for Laos, where the Ministry of Health has classified the project as routine surveillance for the benefit of the country, and removed the requirement for consent. Collaborating research studies included in their own protocol provisions for sample collection procedures and informed consent, compatible with those in the GenRe-Mekong protocol, and obtained ethical approval from both a relevant local ethics review board, and their relevant institutional research ethics committee.

### Decision letter and Author response

Decision letter https://doi.org/10.7554/eLife.62997.sa1
Author response https://doi.org/10.7554/eLife.62997.sa2

## Additional files

### Supplementary files

- Supplementary file 1. Geographical breakdown by year of samples processed by GenRe-Mekong.
- Supplementary file 2. Counts of processed samples, by province/state/division of origin.
- Supplementary file 3. Number of samples carrying mutations in the resistance domains of kelch13 by province/state/division.
- Transparent reporting form

### Data availability

The data used in this paper, including all genotypes, sample metadata and resulting phenotype predictions, are openly available together with detailed methods documentation and details of partner studies, from the article's Resource Page, at http://www.malariagen.net/resource/29.

The following dataset was generated:

| Author(s) | Year | Dataset title | Dataset URL | Database and Identifier |
|---|---|---|---|---|
| Jacob CG, Miotto O | 2021 | Genetic surveillance in the Greater Mekong Subregion and South Asia to support malaria control and elimination: about the data | https://www.malariagen.net/resource/29 | MalariaGEN, 29 |

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
