## [Decision Letter]

**Acceptance summary:**

This work demonstrates the value of malaria genetic surveillance conducted on a large scale. The GenRe-Mekong dataset illuminates heterogeneity in the distribution of drug resistance alleles due to patterns of parasite spread within and across national borders, and underscores the value of data sharing to obtain a regional perspective. While some topics are deserving of deeper treatment, the reviewers accept the authors' perspective that a single manuscript cannot provide a thorough treatment to every dimension of a dataset of this scale.

**Decision letter after peer review:**

Thank you for submitting your article "Genetic surveillance in the Greater Mekong Subregion and South Asia to support malaria control and elimination" for consideration by *eLife*. Your article has been reviewed by 3 peer reviewers, including Daniel E Neafsey as the Reviewing Editor and Reviewer #1, and the evaluation has been overseen by Dominique Soldati-Favre as the Senior Editor. The following individual involved in review of your submission has agreed to reveal their identity: Didier Ménard (Reviewer #2).

The reviewers have discussed the reviews with one another and the Reviewing Editor has drafted this decision to help you prepare a revised submission.

Summary:

This manuscript describes a study (GenRe-Mekong) of 9,623 samples (7,626 ultimately retained for analysis) from malaria-infected patients deriving from eight countries in the Greater Mekong Subregion (GMS). Case studies in particular countries (eg Vietnam, Laos) offer detailed perspective on how resistance alleles may influence outbreaks and case counts locally. This manuscript demonstrates the potential value of systematic application of genomic epidemiological approaches to malaria on a large geographic scale in a region of the world where management of drug resistance is essential for local disease control and elimination efforts. All reviewers agreed this manuscript was well-written, and the work will be of significant interest. Given the large scale of the Genre-Mekong effort, certain aspects of the analysis seem under-developed. The authors may be planning follow up publications to more deeply explore some aspects of the data, but certain features warrant further profiling in the present manuscript. The impact of the manuscript could be enhanced through attention to details that would provide deeper perspective on the data and its relevance to public health. A combined set of suggested revisions from all three reviewers is presented below.

Essential revisions:

1) Given the public health profile of this manuscript, deeper commentary on local actionable decisions that could be informed by the present data should be included.

2) Greater background on the state of malaria in the GMS during the sampling period from the perspective of traditional epidemiological measures and clinical treatment failures would make the paper more accessible to a Broad audience not familiar with malaria in this region.

3) Need for more precise and disciplined language to distinguish 'predicted resistance' from genotypes vs. resistance as measured clinically.

4) More thorough interpretation of the patterns of association between known and candidate resistance markers in this large dataset could be helpful for determining whether any new insights are possible.

5) In general, the issue of distinguishing validated vs. non-validated/candidate resistance mutations needs to be more thoroughly addressed. Some SPOTmalaria markers are validated (evidenced by gene editing) and some are not at all (e.g. exo and arps10, ferredoxin, mdr2, see Table SM1). This need to be clearly stated in the main text. The choice of some molecular markers, which are not validated, has been decided by the authors rather than via recommendation by international organizations. Similarly for Pv molecular markers, it should be stated that none of them have been validated (Table SM2).

6) Attention to additional informative dimensions of the data, such as heterozygosity rate to inform complexity of infection (COI), and interpretation of the spatial and temporal patterns of COI, would further demonstrate the value to public health.

7) It seems that the p-value of an association in the 7K samples used to define imputation rules may give overconfidence in an association, as it is only as representative as the training set. Put another way, there is concern that this might cause regression to the mean, where samples are imputed as having a particular common combination of mutations when in fact this particular sample may simply be different from what we have seen before. In this situation imputation may preclude further investigation of the true genotypes. Perhaps one way around this would be to quantify the degree to which each result relies on imputed values, thereby giving more confidence in more complete results and highlighting areas where more data is needed. The overall degree of imputation in the data (proportion imputed sites) should also be reported in the manuscript.

---

## [Author Response]

Essential revisions:1) Given the public health profile of this manuscript, deeper commentary on local actionable decisions that could be informed by the present data should be included.

We have provided some evidence in the Discussion section, for Vietnam:

“Collaborations with public health organizations have rapidly translated into real impact for malaria control, especially where GenRe-Mekong has been implemented over multiple seasons. Genetic surveillance results were used by the Vietnam NMCP and Ministry of Health in reviews of national drug policy, leading to the replacement of DHA-PPQ with artesunate-pyronaridine as frontline therapy in four provinces. These included the province of Dak Lak, where an early report by GenRe-Mekong in 2018 was the first evidence of ART-R, confirmed by treatment failure data from in vivo therapy efficacy studies (TES) in 2019. In addition, our report of a KEL1/PLA1 outbreak in Gia Lai province has alerted authorities to the need to review the use of DHA-PPQ in that province.”

and also for Laos:

“In Laos, authorities have been equally responsive, using GenRe-Mekong reports in their review of frontline therapy choices: the Ministry of Health opted against adopting the DHA-PPQ ACT based on our evidence of the expansion of resistant strains in the Lower Zone of southern Laos.”

Given the space available, this seemed like adequate coverage of the impact on local decision-making. Perhaps the reviewer meant that we should indicate at the start of the paper the types of application of drug resistance surveillance which would benefit NMCPs. We addressed this by modifying the Introduction section as follows:

“Parasite genetic data is less frequently available, and typically restricted to single genetic variants, or small numbers of sites where quality sample collection protocols could be executed. However, routine mapping of a broad set resistance markers can keep NMCPs abreast of the spread of resistance strains, and help them predict changes in drug efficacy and assess alternative therapies, especially if dense geographical coverage allows mapping of resistance at province or district level. The increased affordability of high-throughput sequencing technologies now offers new opportunities for delivering such knowledge to public health, supporting the optimization of interventions where resources are limited.”

2) Greater background on the state of malaria in the GMS during the sampling period from the perspective of traditional epidemiological measures and clinical treatment failures would make the paper more accessible to a Broad audience not familiar with malaria in this region.

We have expanded the Introduction section with the following:

“This problem is most acutely felt in the Greater Mekong Subregion (GMS), a region that has repeatedly been the origin of drug resistant strains, and in neighbouring countries including Bangladesh and India, where resistance could be imported. The GMS is a region of relatively low endemicity, with entomological inoculation rates 2-3 orders of magnitude lower than in Africa, where the vast majority of cases occur. [Chaumeau et al., Wellcome Open Res 2018; 3: 109, Hay et al., Trans Royal Soc Trop Med Hyg 2000; 94: 113-27.] Infections are most common amongst individuals who work in or live near forests, in remote rural parts of the region. [Cui et al., Acta Trop 2012; 121: 227-39] Since infections are infrequent, a high proportion of individuals in this region are immunologically naïve, and develop symptoms that require treatment when infected. This results in high parasite exposure to drugs, which may be a major evolutionary driving force for the emergence of genetic factors that confer resistance to frontline therapies. [Escalante et al., Trends Parasitol 2009; 25: 557-63] In the past, drug resistance alleles emerged in the GMS and subsequently spread to Africa multiple times, rolling back progress against the disease at the cost of many lives. Currently, global malaria control and elimination strategies depend on the efficacy of artemisinin combination therapies (ACTs) which are the frontline therapy of choice worldwide. Hence, in view of the emergence in the GMS of parasite strains resistant to artemisinin and its ACT partner drug piperaquine, the elimination of *P. falciparum* from this region has become a global health priority.”

3) Need for more precise and disciplined language to distinguish 'predicted resistance' from genotypes vs. resistance as measured clinically.

We have reviewed the whole manuscript to identify all mentions of resistance where there may have been such ambiguities. We made many changes throughout the paper to evidence the predictive nature of the phenotypes (too numerous to list here explicitly). We also modified the “Survey of drug resistance mutations” section of the Results to clarify:

“To bridge this gap, we use genotypes to derive predicted phenotypes based on a set of rules derived from peer-reviewed publications (see Materials and Methods and formal rules definitions available from the article’s Resource Page). These rules predict samples as resistant or sensitive to a particular drug or treatment, or undetermined. Since our procedures do not include the measurement of clinical or in vitro phenotypes, we are only able to predict a drug resistant phenotype based on known associations of certain markers with resistance to certain drugs. Although we report a large catalogue of variations which have been associated with resistance, we do not use all variations to predict resistance. Rather, our predictive rules are conservative and only use markers that have been strongly characterized and validated in published literature and shown to play a crucial role in clinical or in vitro resistance. These critical variants include single nucleotide polymorphisms (SNPs) in genes *kelch13* (resistance to artemisinin), *crt* (chloroquine), *dhfr* (pyrimethamine), *dhps* (sulfadoxine), as well as an amplification breakpoint sequence in *plasmepsin2/3* (marker of resistance to piperaquine). In addition, we report several additional variants found in drug resistance backgrounds but not used to predict resistance, such as mutations in *mdr1* (linked to resistance to multiple drugs), components of the predisposing ART-R background *arps10*, *ferredoxin*, *mdr2*, and the *exo* marker associated with resistance to piperaquine.”

4) More thorough interpretation of the patterns of association between known and candidate resistance markers in this large dataset could be helpful for determining whether any new insights are possible.

This is just one of the many interesting questions that we hope the research community will explore using this dataset. However, it is not within the scope of this paper, which is focussed on the application of genotyping to public health. We had to set some boundaries to the present analysis, and unavoidably, leave some space for future work.

5) In general, the issue of distinguishing validated vs. non-validated/candidate resistance mutations needs to be more thoroughly addressed. Some SPOTmalaria markers are validated (evidenced by gene editing) and some are not at all (e.g. exo and arps10, ferredoxin, mdr2, see Table SM1). This need to be clearly stated in the main text. The choice of some molecular markers, which are not validated, has been decided by the authors rather than via recommendation by international organizations. Similarly for Pv molecular markers, it should be stated that none of them have been validated (Table SM2).

This is a fair comment, and we had assumed that it had been addressed by supplying as supplementary material a whole document detailing explicit rules for predicting drug resistant phenotypes from these markers. In these rules, it is clear that markers such as *arps10*, *ferredoxin* and *mdr2* are not used to predict artemisinin resistance; we genotype and supply them, however, because they have been associated with resistance, and may be (or become) of interest. In the main text, we have now changed Section “Survey of drug resistance mutations” of the main text as follows:

“GenRe-Mekong produces genotypes covering a broad range of known variants associated to drug resistance (Table 2) to support assessment of the spread and risk of drug resistance. [...] Although we report a large catalogue of variations which have been associated with resistance, we do not use all variations to predict resistance. Rather, our predictive rules are conservative and only use markers that have been strongly characterized and validated in published literature and shown to play a crucial role in clinical or in vitro resistance. [...] In addition, we report several additional variants found in drug resistance backgrounds but not used to predict resistance, such as mutations in mdr1 (linked to resistance to multiple drugs), components of the predisposing ART-R background arps10, ferredoxin, mdr2 (Miotto et al., 2015), and the exo marker associated with resistance to piperaquine (Amato et al., 2017).”

It appears to us that the document “The SpotMalaria platform – Technical Notes and Methods” provided in the Supplementary Materials has ample treatment of the roles described in the literature for the markers we genotype, both for Pf and for Pv. Still, we added an introductory paragraph before Tables SM1 and SM2 (Page 4 of that document) as follows:

“All genotyped markers associated with drug resistance are listed in Tables SM1 and SM2 for *P. falciparum* and P. vivax, respectively. A broad spectrum of markers is covered, obtained from scientific literature and public health guidelines, including variants that are validated experimentally, as well as markers associated with resistance but not validated. The prediction of resistant phenotypes requires contextual information about the role of the variant alleles. In the following sections, background information and references are provided, and a separate document formally details the rules which are used to predict drug resistance phenotypes from some of these variants.”

The fact that the Pv markers are just putatively associated with resistance is discussed in the relevant paragraphs of that document.

6) Attention to additional informative dimensions of the data, such as heterozygosity rate to inform complexity of infection (COI), and interpretation of the spatial and temporal patterns of COI, would further demonstrate the value to public health.

This is a reasonable comment, and we can only agree. However, we are still working on the methods of estimation of sample heterozygosity, FWS and COI. At present, we do not feel ready to publish those results, but did not feel we should hold up the present publication for this reason. We will publish an update when that work is complete.

7) It seems that the p-value of an association in the 7K samples used to define imputation rules may give overconfidence in an association, as it is only as representative as the training set. Put another way, there is concern that this might cause regression to the mean, where samples are imputed as having a particular common combination of mutations when in fact this particular sample may simply be different from what we have seen before. In this situation imputation may preclude further investigation of the true genotypes. Perhaps one way around this would be to quantify the degree to which each result relies on imputed values, thereby giving more confidence in more complete results and highlighting areas where more data is needed. The overall degree of imputation in the data (proportion imputed sites) should also be reported in the manuscript.

This is a valid point, but it is useful to frame it in context.

First, we should highlight that imputation rules (see Supplementary materials) are only specified for three genes that possess variants that correlate with mutations conferring drug resistance: *crt* (resistance to chloroquine), *dhfr* (pyrimethamine) and *dhps* (sulfadoxine). No imputation was applied when estimating the prevalence of resistance to artemisinin or piperaquine, which are the most prominent results presented.

Second, the three genes in question have established some common haplotypes in several geographical regions, which have been in circulation for several decades. The high frequency of these haplotypes gives us confidence that they provide reliable imputation variants.

That said, we did compute some statistics to estimate the validity of our imputation rules, by testing whether each imputation rule would predict the correct allele in samples that met the conditions for triggering the rule, but did not require imputation (i.e. the allele to be imputed was present and thus could be compared for agreement). These tests were performed on 21,272 SpotMalaria samples, including all samples in the GenRe-Mekong study and several others from other studies using SpotMalaria. We found the following:

– For all rules (a total of 70,596 triggered cases), we found agreement in 99.1% of cases.

– When stratifying by gene, agreement was >99.9% for the *crt* and *dhfr* genes, while *dhps* (involved in sulfadoxine resistance) had 96.7% agreement.

– When considering only rules that affect the phenotype prediction, agreement was still >99.9% for the *crt* and *dhfr* genes, while *dhps* agreement was 95.4%.

– Even if we take imputation rule disagreements to represent a rule error rate, this does not equate to a phenotype prediction error rate, since rules are only triggered for a minority of samples.

The worst-performing rule for *dhps* had a disagreement rate of 8.9%, and was used to impute 303 samples out of 21,272 (i.e. 1.4% of samples). Thus, we estimate it could introduce phenotype prediction error of 0.089 0.014, i.e. around 0.1%. The second-worst rule had a disagreement rate of 3.2% and was used on 102 samples, introducing an estimated phenotype prediction error of around 0.02%.

After reviewing the above results, such minimal effects on phenotype predictions were deemed acceptable.